# A Method for Exploring and Analyzing Spatiotemporal Patterns of Traffic Congestion in Expressway Networks Based on Origin–Destination Data

Hong Gao [1], Zhenjun Yan [1], Xu Hu [1], Zhaoyuan Yu [1,2,3], Wen Luo [1,2,3,*], Linwang Yuan [1,2,3] and Jiyi Zhang [4,5]

[1]  School of Geography, Nanjing Normal University, Nanjing 210023, China; 181301015@njnu.edu.cn (H.G.); 191301026@njnu.edu.cn (Z.Y.); 191302068@njnu.edu.cn (X.H.); yuzhaoyuan@njnu.edu.cn (Z.Y.); yuanlinwang@njnu.edu.cn (L.Y.)

[2]  Key Laboratory of Virtual Geographic Environment, Nanjing Normal University, Ministry of Education, Nanjing 210023, China

[3]  Jiangsu Center for Collaborative Innovation in Geographical Information Resource Development and Application, Nanjing 210023, China

[4]  College of Geographic Science, Nantong University, Nantong 226019, China; zhangjiyi@ntu.edu.cn

[5]  Department of Geographic Information Science, Chuzhou University, Chuzhou 239000, China

*  Correspondence: 09415@njnu.edu.cn

**Abstract:** Traffic congestion in expressway networks has a strong negative influence on regional development. Understanding the spatiotemporal patterns of traffic congestion in expressway networks is critical for improving the exchange of products in regional production and promoting regional economic development. Nevertheless, existing studies pay less attention to these spatiotemporal patterns of traffic congestion. Considering that Origin–Destination (OD) data are available for the recorded spatial movements of vehicles in expressways, this study proposes a method with which to explore traffic congestion at the level of road segments in the regional expressway network, the mainstream of driving behaviors, and traffic regulations. Methods for analyzing spatial disparity and temporal changes in traffic congestion in expressway networks are also put forward in this paper. The empirical results show that the proposed methods could detect road segments where traffic congestion happens, and then uncover temporal patterns of several congested locations and spatial patterns of road segments with frequent congestion. These spatiotemporal patterns of traffic congestion could be in accord with the actual situation. This study provides a new approach to investigating traffic congestion in expressway networks based on low-cost data, which might be helpful for optimizing expressway network planning and promoting balanced regional development.

**Keywords:** traffic congestion; spatiotemporal patterns; road segment; expressway network; origin–destination data

## 1. Introduction

Expressways are an important element influencing regional economies [1–4] because they are the main method for the rapid transportation of passengers and merchandises between different cities. Unimpeded transportation in expressway systems would accelerate the exchange of products or raw materials for regional production, which could promote regional economic development. However, the traffic congestion in expressway networks would inevitably decrease the transporting efficiency and increase the time cost [5–7], which might influence regional economic development [8,9]. Understanding where traffic congestion occurs most frequently and the spatiotemporal patterns of these places in expressway networks are important to guide traffic flow management, optimize expressway network planning and promote balanced regional development. Therefore, it is critical to explore and analyze spatiotemporal patterns of traffic congestion in regional expressway networks.

However, existing studies pay less attention to the spatiotemporal patterns of traffic congestion for regional expressway networks. Instead, they mainly focus on the traffic congestion of urban roads and have developed various methods to identify [10–16], predict [17–21], and analyze [22–29] urban traffic congestion. Compared with urban traffic congestion, there are fewer studies that focus on traffic congestion in expressways. The existing studies are based on different data and methods to investigate traffic congestion in expressways [30–40] and analyze the characteristics of traffic congestion in expressways [41–45]. The summary of representative studies about traffic congestion in expressways is listed in Table 1. We found that existing studies about traffic congestion in expressways ignore the traffic congestion from the expressway network perspective. These studies lack analysis of the spatiotemporal patterns of traffic congestion in regional expressway networks, while such patterns are critical to improving regional planning, commodity circulation, and regional economic development.

**Table 1.** The summary of representative studies about traffic congestion in expressway.

| Data Source or Type | Model | Application | Contribution |
| --- | --- | --- | --- |
| Traffic detector | A fusion method [31]; Fuzzy inference approach [32]; A practice-ready method [33]; Bayesian robust tensor factorization model [34] | Segment of an urban expressway | Location positioning of traffic congestion |
| Vehicular network | A traffic congestion detection and information dissemination scheme [35] | Segment of an urban expressway | Ensure the accuracy of estimating congestion level |
| Vehicle trajectories | Image processing method [37]; Neural networks model [38] | Segment of expressway | Analyze the spatial-temporal distribution of traffic congestion |
| Traffic volume, speed, and travel time | Process simulation model [39]; Cell transmission model [40] | Segment of an urban expressway | Provide the underlying insights of traffic congestion mechanism |
| Traffic big data | Flow-speed fundamental diagram [45] | Ring road of an urban expressway | Identify the pattern of the recurrent traffic congestions |
| Remote sensing data | Federated learning [13] | Road including expressway | Detect the spatial range of traffic congestion and ensure the data privacy |

One of the main reasons for the lack of studies on the traffic congestion of the expressway networks might be that it is difficult or expensive for the current data types to cover the whole expressway network. Traffic detectors and vehicle networks are expensive and only cover limited areas [24]. Obtaining trajectory data or big traffic data requires installing special devices or application software in each vehicle [24], which might be difficult to implement for the numerous vehicles in an expressway network. In any case, it would be difficult to obtain data for expressway networks where there is no (or poor) GPS signal. Remote sensing data might not provide time-scale data of sufficient resolution (i.e., hourly) for detecting traffic congestion in expressway networks. Additionally, remote sensing data cannot be collected when visibility is poor. Therefore, it is necessary to develop a new method that could provide a low-cost data source with which to explore the traffic congestion in expressway networks.

Origin–Destination (OD) data are an important data source for recording the spatial movements of vehicles that are relatively easy to obtain for expressway networks. For example, they can be collected by a toll system in the expressway networks. However, OD data emphasizes the origin and the destination in the spatial movement while ignoring the actual trajectory [46–48], making it difficult to analyze the traffic congestion in an expressway network using OD data. However, after recovering driving routes for OD data

and analyzing the overall characteristics of these routes, it might be practicable to use OD data to analyze traffic congestion in regional expressway networks.

In this study, we developed a method for exploring and analyzing spatiotemporal patterns of traffic congestion in regional expressway networks. The proposed method screens abnormal OD records with a lower-than-normal average speed to determine congested routes and select congested road segments. Then, the method characterizes and analyzes temporal changes and spatial disparity in traffic congestion. The main focus of this study was to provide a new approach for uncovering traffic congestion spatiotemporal patterns in regional expressway networks, which are generally ignored by many existing studies. The developed method could help policymakers and regional managers to improve regional expressway network planning and economic development.

The remainder of this paper is organized as follows. Section 2 presents the basic idea of the proposed method; Section 3 introduces the method; Section 4 contains a case study; Section 5 provides the discussions and conclusions about this study.

## 2. Basic Idea

### 2.1. Related Definitions

**Definition 1.** *Traffic congestion. Traffic congestion is the condition that causes the total travel time of a vehicle on an expressway to exceed that required for normal driving [49]. Traffic congestion makes the average speed of a vehicle on an expressway lower than the minimum speed in a normal situation.*

**Definition 2.** *Expressway network. A toll road network system for rapid transportation. Expressway networks are specific stations (toll stations) letting vehicles enter or exit. Vehicles cannot drive in or drive out of expressways arbitrarily, only at these stations. Expressway networks have detailed regulations about vehicle speed, including maximum and minimum speed limits.*

**Definition 3.** *Origin–Destination (OD) data. A data type recording the spatial movements of vehicles on an expressway. Each OD record emphasizes the origin location (station) and the destination location (another station) for each spatial movement of vehicles [50]. OD records include two time stamps at the origin location and the destination location.*

**Definition 4.** *Road segment. Expressway stations (entrance and export) and road junctions are breakpoints that divide expressway networks into many road segments. Road segments were the basic unit for measuring spatial distributions of traffic congestion in this study.*

**Definition 5.** *Driving route. Each OD record has a corresponding driving route in the expressway network, from the origin location to the destination location. A series of ordered road segments represent the driving route in OD records. OD records with the same origin and destination (at different times) may have the same driving route.*

**Definition 6.** *Abnormal OD (AOD). An abnormal OD record is one with a long overall travel time on the expressway making its average speed lower than the minimum speed of the normal situation.*

**Definition 7.** *Congested route (CR). A driving route between origin and destination where traffic congestion occurs.*

**Definition 8.** *Congested road segment (CRS). A road segment in a congested route.*

### 2.2. Framework to Explore Traffic Congestion in Expressway Network

Expressway OD data consist of massive OD records, and each of them shows when a vehicle enters and exits an expressway. Traffic congestion would increase the total travel time of a vehicle on an expressway, deduced from the start and end times in the OD

record. Thus, OD data can provide important information about travel time for analyzing traffic congestion.

Toll expressway networks are closed systems in which vehicles must drive in or drive out at toll stations. The vehicle can only move through the expressway network between the origin station and the destination station. Generally, there is more than one route in an expressway network between two locations. The length of a route is positively correlated to the transport cost and time. The cost and efficiency are important factors for drivers to select a route.

In most cases, drivers would choose the shortest route in the expressway network between their origins and destinations, improving efficiency and reducing costs. Therefore, the driving routes of OD data could be recovered by using the shortest path-searching algorithm. Besides, other situations might increase the travel time of the OD record, other than traffic congestion, such as drivers not choosing the shortest route. These situations generally involve a small proportion of vehicles on the same route. However, traffic congestion would influence a large proportion of vehicles in a short time, increasing their travel time; thus, traffic congestion could be distinguished.

Based on the above, we could design a method to utilize OD data for exploring traffic congestion in expressway networks. The method is illustrated in Figure 1. The expressway network and OD data are in Figure 1A. The toll station is the fixed entrance or exit of an expressway network, and between two traffic stations, the vehicles can only drive along the expressway, and driving routes could be recovered from the OD data. The congested route is distinguished according to the relationship between abnormal and all OD records (Figure 1B). The congested road segment is selected from each congested route (Figure 1C), and the congested road segments from different congested routes are overlaid on the expressway network to obtain the spatial distribution of traffic congestion at the level of the road segment (Figure 1D). The spatial overlay of three congested road segments is shown in Figure 1C,D. According to the spatial overlay of congested road segments in different time intervals, we would further analyze the spatiotemporal patterns of traffic congestion in the expressway network.

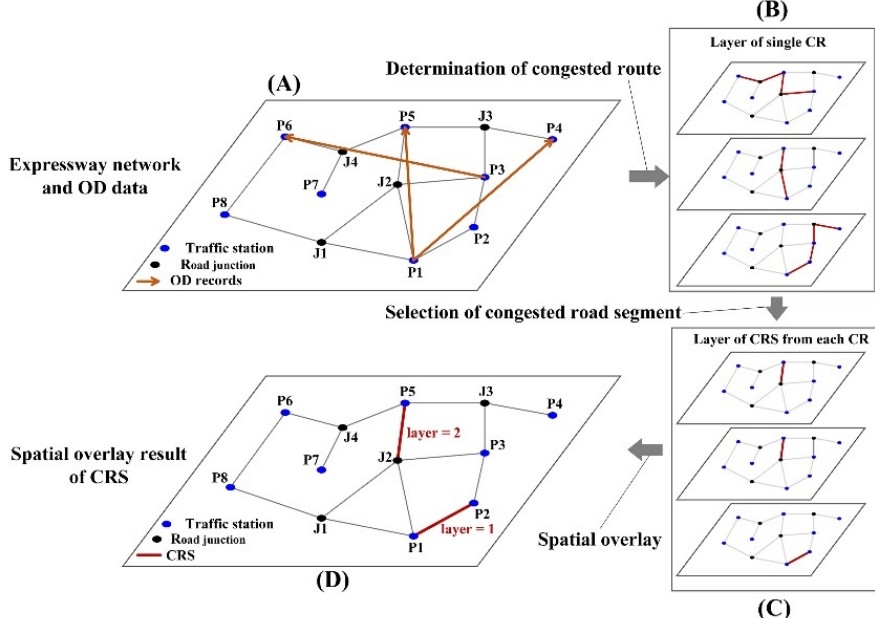

**Figure 1.** Illustration of the workflow in the basic idea. (**A**) the example of expressway network and OD data, (**B**) the spatial layer of single congested route, (**C**) the spatial layer of congested road segment selected from each congested route, (**D**) the result of spatial overlay of congested road segments.

## 3. Methods

### 3.1. Detect Congested Road Segments in Expressway Network

In this section, we introduce how to detect congested road segments in expressway networks using OD data. The driving routes of OD data are inferred using mainstream driving behaviors, abnormal OD records are screened by considering the minimum speed limit in expressway regulations, and the congested path and road segments are determined according to the proportion of abnormal OD records in all OD records.

#### 3.1.1. Recover the Driving Routes for OD Data

The OD data include the origin and the destination for each spatial movement of the vehicle by the expressway but ignore the vehicle trajectory. It is essential to recover the driving routes for OD data to determine where traffic congestion occurs.

Expressway networks with toll systems are a closed system; vehicles cannot drive in or out of an expressway arbitrarily, except at specific stations. After entering the expressway, the vehicle can only drive along the expressway network until exiting. Most drivers would choose the shortest route between origin and destination to promote transport efficiency as it decreases driving costs (time and economic costs). Therefore, the mainstream driving behavior is the driver choosing the shortest route between the origin and the destination. Accordingly, the shortest route between the origin and destination in the OD data could be inferred as the driving route. This driving route could be determined in an expressway network by an optimal path-searching algorithm, such as the Dijkstra algorithm [51]. The driving route in the OD record consists of serial road segments in order, meaning the vehicle passed these road segments in turn.

It should be noted that the shortest route is available for most OD records. However, some vehicles might not use the shortest route, such as drivers making a detour. The corresponding strategy should be designed to eliminate the influence of other reasons on detecting congested road segments.

Traffic congestion would block all vehicles, increase their total travel time, and decrease their average speeds while driving, generating many abnormal OD records in a short time. Therefore, traffic congestion represents a high proportion of abnormal OD records on the same driving route. However, the abnormal OD records caused by other reasons, such as detours, may occupy a limited proportion for the same route. Therefore, the influence of other situations on detecting traffic congestion could be overcome by analyzing the proportion of abnormal OD records to OD records for each route.

#### 3.1.2. Determination of Congested Routes in Expressway Network

(1) Screen the Abnormal OD Records (AOD)

An abnormal OD record has a lower average speed than the minimum speed of normal. Thus, the average speed of each OD record needs to be calculated to screen the abnormal OD records using a speed threshold. The average speed of an OD record is calculated by the length of the recovered driving route and the travel time calculated by the start time and the end time. The speed threshold could be set by considering expressway regulations.

Expressways generally have minimum speed limits to promote transport efficiency. It is prohibited for a vehicle to drive in an expressway slower than the minimum limit. However, traffic congestion would unavoidably increase vehicle total travel time, reducing their average speed below the minimum limit. Therefore, a lower speed than the minimum limit could be the threshold for selecting abnormal OD records. The algorithm for abnormal OD record screening is in Algorithm 1.

---

**Algorithm 1:** Algorithm to screen AOD

---

**Input:** $T = \{ T_i | 1 \leq i \leq m \} = \{ Origin_i, Time_{oi}, Destination_i, Time_{di} \}$, expressway OD data; $N = \{ V, E \}$, expressway network, where $V = \{ V_p \}$ are breakpoints (road junctions and traffic stations), and $E = \{ E_q \}$ are the road links (with length); $s_0$ is the speed threshold from expressway regulations.

**Output:** $AT = \{ AT_j | 1 \leq j \leq n \} = \{ Origin_j, Time_{oj}, Destination_j, Time_{dj} \}$ gives abnormal OD records.

**Steps:**

1. Take a single OD record $T_i$ from OD data $T$;
2. Calculate the travel time $\Delta t_i$ of OD record $T_i$, which equals $Time_{di} - Time_{oi}$;
3. Recover the driving route $r_i$ between $Origin_i$ and $Destination_i$ in a road network $N$, using the Dijkstra algorithm;
4. Calculate the length of the route $r_i$, which is $L_i$;
5. Calculate the average driving speed $s_i$, which is calculated by $L_i \div \Delta t_i$;
6. Compare average speed $s_i$ and speed threshold $s_0$:
(1) If $s_i \geq s_0$, continue;
(2) If $s_i < s_0$, add this OD record $T_i$ into the dataset of abnormal OD records $A\ T$, $AT_j = T_i$, and then $j = j + 1$;
7. Judge if $i = m$:
(1) No, $i = i + 1$, and back to step 1;
(2) Yes, return $AT = \{ AT_j \}$.

---

(2) Determine the congested routes (CR)

Traffic congestion influences all vehicles; when traffic congestion occurs, it would cause many abnormal OD records in a short time (i.e., an hour). Based on these analyses, we could use the proportion of abnormal OD records in all OD records over a short time to judge whether a route is experiencing traffic congestion. This proportion is calculated by Equation (1):

$$P^r_{AOD} = \frac{\sum\limits_{j=1} AOD^r_j}{\sum\limits_{i=1} OD^r_i} \times 100\% \tag{1}$$

where $AOD^r_j$ is the $j$-th abnormal OD record in the route ($r$), and $OD^r_i$ is the $i$-th OD record in $r$.

The proportion is highly influenced by the travel times in OD data; if the travel time is longer, then the average speed is lower. If the quantity of OD records with a long travel time is large, the quantity of abnormal OD records may increase. Thus, this proportion would increase. This proportion would be calculated for different time intervals to determine the congested route in the expressway network in each time interval.

In this study, the route with many abnormal OD records (more than 80%) would be determined as a congested route. The large proportion of abnormal OD records could indicate that plenty of vehicles had a longer travel time than the usual in a route, indicating traffic congestion.

Generally, traffic congestion would influence many vehicles over a short time, increasing their travel time and decreasing average speeds; thus, causing many abnormal OD records. However, other situations, which may also increase travel time and abnormal OD records, rarely appear for vehicles simultaneously over a short time. Labeling routes with large proportions of abnormal OD records as congested could reduce the influence of other situations on detecting traffic congestion. Therefore, the proposed method could distinguish traffic congestion from other routine situations.

3.1.3. Selection of Congested Road Segments (CRS)

(1) Select Congested Road Segment in a Congested Route

A congested route in an expressway network might include several road segments, as a congested route is too vague to reflect the spatial distribution of traffic congestion in the expressway network. Further determining the road segment where traffic congestion occurs in each congested route by removing road segments with no congestion could also be performed using the proportion of abnormal OD records.

When there is no traffic congestion in a road segment, all vehicles drive normally, which would generate low abnormal OD records in this road segment. Therefore, the normal road segments could be identified by small proportions of abnormal OD records. Finally, after removing normal road segments, the congested road segment could be selected from a congested route. Algorithm 2 shows the algorithm to select congested road segments.

---

**Algorithm 2:** Algorithm for determining congested road segment

---

**Input:** $CR = \{rs_1, rs_2, \cdots, rs_n\}$, a congested route that is consist of $n$ road segment; $OD = \{od_i\}$, all OD records; $AOD = \{aod_i\}$, abnormal OD records.
**Output:** $CRS = \{crs_i\}$, congested road segment.
**Steps:**
    1. Take road segment $rs_k$ from CR.
    2. Take all abnormal OD records whose route is $rs_k$, denoted as $\{aod_i^k\}$.
    3. Tale all OD records whose route is $rs_k$, denoted as $\{od_i^k\}$.
    4. Calculate the proportion of abnormal OD records in $rs_k$, using Equation (1), denoted as $p_k$.
    5. If $p_k$ is smaller than the proportion threshold, remove $rs_k$ from this congested route.
    6. If $k = n$:
      (1) No, $k = k + 1$, back to step 1;
      (2) Yes, end.

---

(2) Classify traffic condition for each road segment by spatial overlay.

By overlaying the layer of different congested road segments on an expressway network, the traffic condition of each road segment in the expressway network could be characterized (Figure 1). A road segment might be overlaid by multiple congested road segments, so the traffic congestion at the current road segment would have influenced more than one route, reflecting the intensity of this traffic congestion. Therefore, this method would use the number of congested road segments calculated by spatial overlay to classify the traffic condition for each road segment of the expressway network.

The places with no congested road segment overlay reflect not being affected by traffic congestion. One layer of congested road segment indicates traffic congestion, while more than one congested road segment indicates more serious traffic congestion. Table 2 shows the classification of traffic conditions according to the layer number of congested road segments.

**Table 2.** Classification of traffic condition by spatial overlay of CRS.

| Number of Layers | 0 | 1 | 2–3 | >3 |
|---|---|---|---|---|
| Traffic condition | Smooth | Mild congestion | Moderate congestion | Serious congestion |

### 3.2. Characterizing the Spatiotemporal Patterns of Traffic Congestion in an Expressway Network

This section mainly introduces how to classify traffic congestion at a road segment and construct the features of traffic congestion according to congested road segments, to analyze the temporal changes and spatial disparity of traffic congestion in expressway networks.

#### 3.2.1. Determine the Direction of Traffic Congestion at a Road Segment

It should be noted that traffic congestion has a direction attribute. For the same expressway, one direction may be congested. The direction of traffic congestion is helpful to understand its adverse impact on decreasing transport efficiency. In this method, traffic congestion is detected at the road segment level. According to the driving direction of vehicles passing through the congested road segment, four directions of traffic congestion (eastward, westward, northward, and southward) are classified in this method.

For example, there is a congested road segment with two vertexes, which is $L_1(x_1, y_1)$ and $L_2(x_2, y_2)$. If $|x_1 - x_2| > |y_1 - y_2|$, the dominant tendency of this road segment lies from east to west. Thus, the direction of congestion at this road segment is either eastward

or westward. Specifically, when a vehicle drives from $L_1$ to $L_2$, $x_2 > x_1$ traffic congestion is eastward and when $x_2 < x_1$ traffic congestion is westward.

If $|x_1 - x_2| < |y_1 - y_2|$, it indicates that the dominant tendency of this road segment lies from south to north, and when a vehicle drives from $L_1$ to $L_2$, $y_2 < y_1$ it means that the traffic congestion is southward, and $y_2 > y_1$ means that the traffic congestion is northward.

The criterion to classify the direction of traffic congestion at road segments is shown in Table 3.

**Table 3.** The criterion to classify direction of traffic congestion at a road segment.

| Tendency of Congested Road Segment | Driving Direction | Congested Direction |
|---|---|---|
| $|x_1 - x_2| \geq |y_1 - y_2|$  or | From $L_1$ to $L_2$ | Eastward congestion |
| | From $L_2$ to $L_1$ | Westward congestion |
| $|x_1 - x_2| < |y_1 - y_2|$  or | From $L_1$ to $L_2$ | Southward congestion |
| | From $L_2$ to $L_1$ | Northward congestion |

### 3.2.2. Temporal and Spatial Characteristics of Traffic Congestion

To analyze the spatiotemporal patterns of traffic congestion in an expressway network, one hour was set as the time interval, and OD data in the same hour was used to select congested road segments. According to congested road segments in the serial hours of one year, the temporal changes and spatial disparity of traffic congestion in the expressway network could be analyzed.

(1) Characterize the temporal changes of traffic congestion.

**Definition 9.** *Number of congested locations. The number of congested locations indicates how many locations in the expressway network happened traffic congestion inner a day. Because congested road segments are detected each hour, the average value of numbers of congested locations in one day is calculated to illustrate the number of congested locations in a day.*

A congested road segment in one hour means one congested location in this hour. This process could be expressed as Equation (2):

$$m_h^r = \begin{cases} 1, & \text{if } r \text{ is CRS in the hour } h \\ 0, & \text{if } r \text{ is not CRS in the hour } h \end{cases} \tag{2}$$

where $h$ is the hour, and $r$ is a road segment in the expressway network. $m_h^r$ is the number of congested locations in a road segment $r$ in the hour $h$.

The number of congested locations in an expressway network in the same hour is added to get the number of congested locations in an hour. This process is expressed by Equation (3):

$$T_h = \sum_{r=1}^{r=R} m_h^r \tag{3}$$

where $h$ is the hour, and $r$ is a road segment in the expressway network. $R$ is the total number of road segments in an expressway network. $T_h$ is the number of congested locations in an expressway network in the hour $h$.

After that, the average number of congested locations in 24 h of one day is calculated to characterize the number of congested locations for this day. This process is expressed by Equation (4):

$$T_d = \frac{1}{24} \sum_{h=1}^{h=24} T_h \tag{4}$$

where $h$ is the hour, $T_h$ is the number of congested locations in an expressway network in an hour $h$, and $T_d$ is the number of congested locations in an expressway network in a day $d$.

Based on congested road segments in each hour, time series of congested locations in a day could be generated to analyze the temporal changes of traffic congestion in an expressway network. The algorithm for generating the time series of congested locations is illustrated in Algorithm 3.

---

**Algorithm 3:** Algorithm for generating time series of several congested locations

---

**Input:** $CRS = \{crs_i\}$, congested road segments in each hour of one year; $RS = \{r_i | 1 \leq i \leq R\}$, all road segments in an expressway network.

**Output:** $TS = \{T_d\}$, time series of the number of congested locations each day.

Steps:

1. Take one day $d$ in a year;
2. Take one hour $h$ in the day $d$;
3. Take all congested road segments in an hour $h$, denoted as $\left\{ crs_i^h \right\}$;
4. Take a road segment $r$ from the expressway network, and use Equation (2) to calculate the number of congested locations in $r$, to get $m_h^r$;
5. Judge if $r = R$:
   (1) Yes, use Equation (3) to to calculate the number of congested locations in an hour $h$, to get $T_h$;
   (2) No, $r = r + 1$, back to step 4;
6. Judge if $h = 24$:
   (1) Yes, use Equation (4) to calculate the number of congested locations in the day $d$, to get $T_d$;
   (2) No, $h = h + 1$, back to step 2;
7. Judge if $d = 365$:
   (1) Yes, end this procedure;
   (2) No, $d = d + 1$, back to step 1.

---

(2) Characterize the spatial disparity of traffic congestion

**Definition 10.** *Frequency of traffic congestion. How many times traffic congestion had happened in a road segment in a year.*

A congested road segment in one hour is one instance of traffic congestion for this road segment. This process could be expressed by Equation (5):

$$n_r^h = \begin{cases} 1, & \text{if } r \text{ is CRS in the hour } h \\ 0, & \text{if } r \text{ is not CRS in the hour } h \end{cases} \tag{5}$$

where $h$ is the hour, and $r$ is a road segment in an expressway network. $n_r^h$ is the time of traffic congestion for a road segment $r$ in an hour $h$.

For a single road segment, the traffic congestion across all hours of a year is summed to calculate the frequency of traffic congestion, as expressed by Equation (6):

$$R_r = \sum_{h=1}^{h=H} n_r^h \tag{6}$$

where $h$ is hour, H is the total number of hours in a year. $n_r^h$ is the time of traffic congestion for road segment $r$ in hour $h$. $R_r$ is the frequency of traffic congestion for a road segment $r$ in a year.

The frequency of traffic congestion could be calculated for each road segment in an expressway network. Algorithm 4 illustrates how to calculate the frequency of traffic congestion for each road segment of an expressway network.

---

**Algorithm 4**: Algorithm to calculate the frequency of traffic congestion for every road segment

---

**Input:** $CRS = \{crs_i\}$, congested road segments in each hour of one year; $RS = \{r_i | 1 \leq i \leq R\}$, all road segments in an expressway network; $H$, the total number of hours in a year.
**Output:** $F = \{f_r\}$, frequency of traffic congestion for each road segment.
**Steps:**
    1. Take a road segment $r$;
    2. Take an hour $h$;
    3. Use Equation (5) to calculate the times of traffic congestion for $r$ in $h$, and get $n_r^h$;
    4. Judge if $h = H$:
        (1) Yes, use Equation (6) to calculate the frequency of traffic congestion for $r$, and get $f_r$;
        (2) No, $h = h + 1$, back to step 2.
    5. Judge if $r = R$:
        (1) Yes, end this procedure;
        (2) No, $r = r + 1$, back to step 1.

---

## 4. Case Study

### 4.1. Study Region and Data

(1) Study region

This study took the regional expressway system among 25 cities located in Jiangsu province, southeastern China, as the study area. The study area is a developed region in China, with 230 toll stations in the study region were set to let vehicles enter or exit the expressway system and collect OD records for each vehicle. The study area had an expressway network with a total length of ~3008 km. The density of the expressway network was 700 m per square kilometer in the study region. The spatial location of the study region, expressway network, and toll stations are shown in Figure 2.

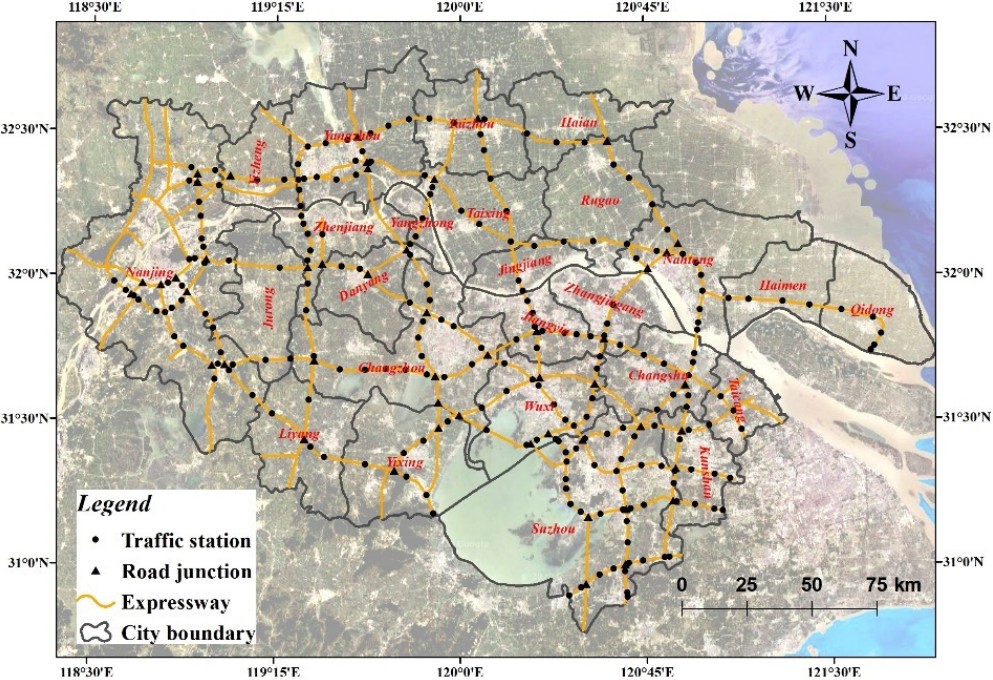

**Figure 2.** The spatial location of the study region and expressway network. The background is a remote sensing image from Google.

(2) OD data in expressway network

The OD data about passenger vehicles (less than seven seats) in 2015 (except for several days when the expressway is free of charge to travel) were used to investigate traffic congestion in the study region. These OD data were from the toll collection system installed in the fixed entrance and export of the expressway [50,52]. Each vehicle is required to pay a fee to use the expressway for transporting. The toll collection system recorded when and where the vehicle drove in and out of the expressway system to determine user charges. Thus, the toll collection system generated the OD data of vehicle movement.

Each OD record had the attributes of record ID, enter time, enter station time, exit time, and exit station time. The examples of expressway OD records used in this study are in Table 4, where the entry station and the exit station are the code number of traffic stations. The same traffic station has the same unique code number and the same spatial coordinates in the expressway network. The time stamps of entering and exiting are accurate to seconds, which are used to calculate the travel time of each vehicle.

**Table 4.** Example of expressway OD records in this study.

| Record ID | Enter Time | Enter Station | Exit Time | Exit Station |
|---|---|---|---|---|
| 12109 | 0:07:14 1 January 2015 | 2090003 | 1 January 2015 0:23:12 | 1700202 |
| 13347 | 0:07:47 1 January 2015 | 2110004 | 1 January 2015 0:17:03 | 2110001 |
| 12058 | 12:15:04 1 January 2015 | 2060002 | 1 January 2015 12:26:46 | 2060005 |
| 2601 | 7:36:08 1 January 2015 | 1650005 | 1 January 2015 7:49:36 | 1650004 |
| 12152 | 8:14:31 1 January 2015 | 1700101 | 1 January 2015 9:35:37 | 2090001 |

(3) Parameters in the case study

In this case study, all the OD data in 2015 were grouped by hours, and the congested road segments every hour over the year were detected. The minimum speed limit of the expressway is 60 km/h, so we set 50 km/h as the threshold to screen for abnormal OD records, and 80% was set as the proportion of abnormal OD records in OD records to determine a congested route, and 40% was set as the proportion threshold for selecting congested road segments in congested routes.

*4.2. The Congested Road Segment Selected by the Proposed Method*

This study detected congested road segments each hour over a year and classified the direction of congestion at the road segment level. Figure 3 shows the spatiotemporal distribution of congested road segments in different directions, whose time range was the first week of the year (1 January to 7 January). The days from 1 January to 3 January were New Year's Day Holiday, and the days from 4 January to 7 January were working days.

During the selected days, it was found that more road segments had southward and northward congestion than eastward and westward congestion, which could reflect the inconvenience of transporting between the south and the north in the study region. It was found that traffic congestion occurs near 10:00 on 1 January. The possible reason might be that it is the first day of the holiday when many people drive to another city, which may cause traffic congestion. At 17:00 on 3 January, there were also many congested road segments, mainly with southward congestion, because it is the last day of the holiday when people return to their original cities.

It was noticed that there were more road segments with westward and northward congestion on 1 January, because the cities of Suzhou, Wuxi, Changshu, Taicang, and Kunshan (near Shanghai) were southeast of the study region, which have developed economies and large populations. Thus, these regions would have numerous vehicles using the expressway to get to other cities for vacations, which could cause westward and northward congestions. When the holiday ends on 3 January, many vehicles return southeast, which might cause southward and eastward congestions in the study region. This phenomenon could reflect the relationships between the economy and traffic flow, demonstrating the effectiveness and rationality of research results.

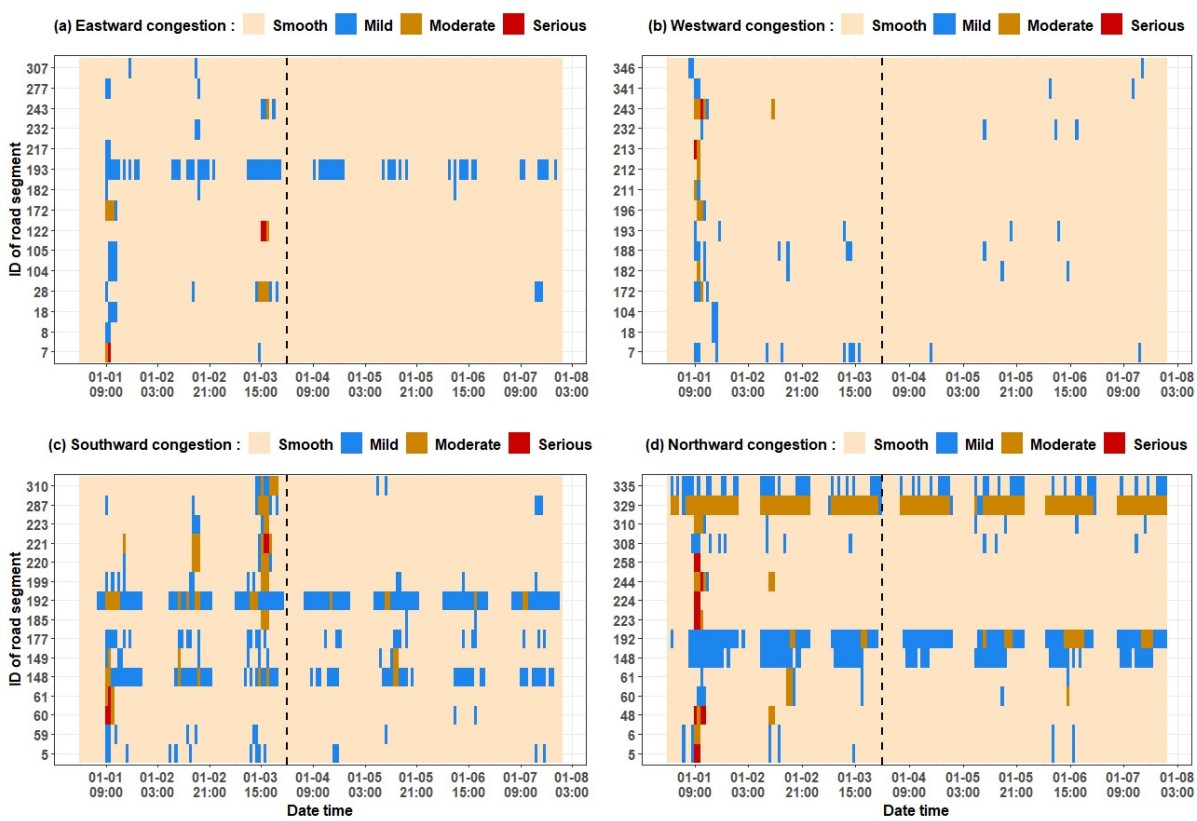

**Figure 3.** The spatiotemporal distribution of congested road segments.

*4.3. Temporal Changes of Traffic Congestion in Expressway Network*

The number of congested locations for each day in a year was calculated, and the time series of congested locations were generated to analyze the temporal changes in traffic congestion in the expressway network. Figure 4 shows the time series of congested locations in different directions in 2015.

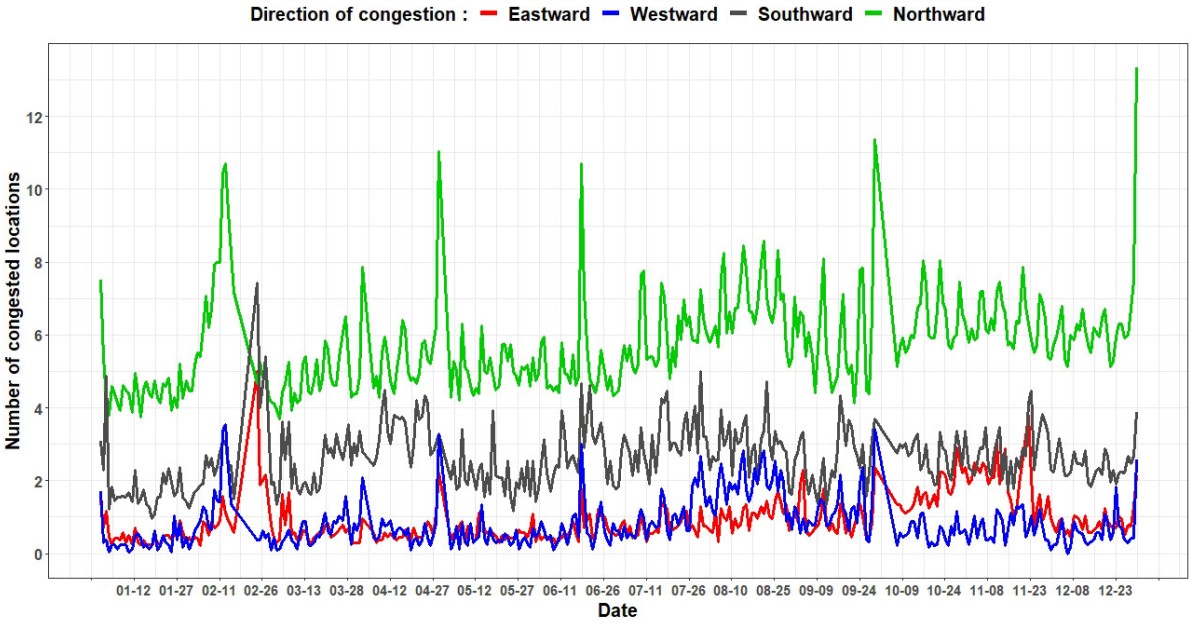

**Figure 4.** The time series of congested locations in different directions.

It was found that there was higher northward and southward congestion than in the other two directions nearly every day. A possible reason might be that the Yangtze River crosses the study region from the east to west and hinders the transport between the south and the north, causing more southward and northward congestions. In any case, the northward congestion occurred in most places in the expressway each day because the cities south of the study region have a developed manufacturing industry, such as Changshu, Suzhou, and Taicang. A large volume of industrial products in these cities would be transported by the expressway to northern cities via numerous vehicles and might cause northward congestion. The eastward and westward congestion occurred in a small number of locations most days because there were more road segments between the east and the west in the study region than between the north and south. Another possible reason is that there was no natural hindrance between the east and the west in the study region.

Many days in January had a smaller number of congested locations than in other months. However, there were more places with traffic congestion in the days of August. There were several obvious peak congested locations over the year, which could be related to the holiday period, as people would choose expressway to travel, causing traffic congestion in many places of the expressway network.

### 4.4. Spatial Disparity of Traffic Congestion in Expressway Network

The frequency of traffic congestion was calculated for each road segment in the expressway network. Figure 5 shows the spatial distribution of road segments with a high frequency of traffic congestion in different directions in 2015.

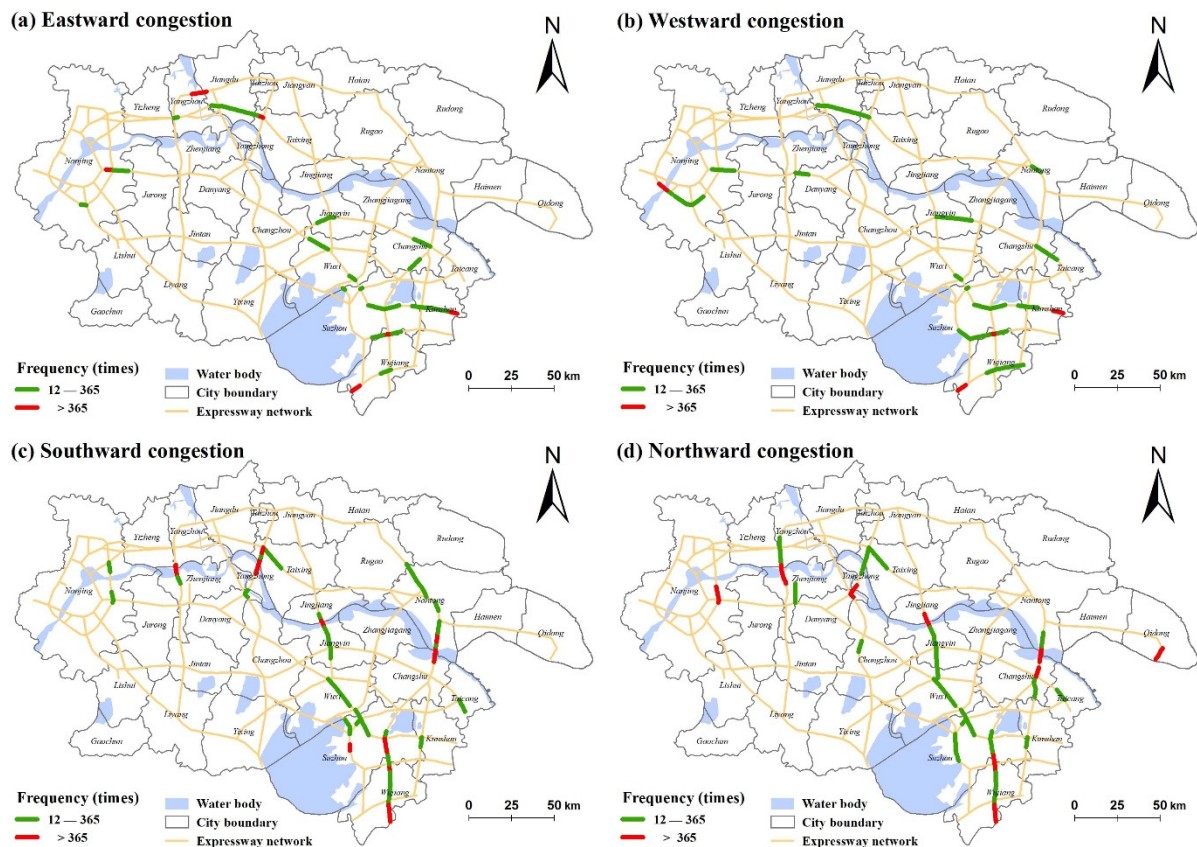

**Figure 5.** The spatial distribution of road segments with frequent congestion.

For eastward and westward directions, there are fewer road segments with less congestion. The road segments with frequent eastward congestion were mainly located in Nanjing, Yangzhou, Kunshan, and Wujiang, while Nanjing, Kunshan, and Wujiang also had road segments with frequent westward congestion. There were more road segments with frequent congestion in the southward and northward directions than that in other directions. The road segments with frequent southward congestion were mainly located in Zhenjiang, Yangzhong, Changshu, Suzhou, and Wujiang. In the northward direction, the road segments with frequent congestion included Nanjing, Zhenjiang, Yangzhong, Jingjiang, Changshu, Qidong, and Wujiang.

The spatial distribution findings of the frequency of traffic congestion might be helpful when planning expressway networks. According to the current results, it might be beneficial to build an expressway between Nanjing and Jurong and between Yangzhou and Taixing because the existing expressway in these regions experienced frequent eastward and westward congestion. Seeing that the expressway in Wuxi had the most congestion for both southward and northward directions, it could be improved by building another expressway northward. Finally, many road segments with frequent congestion were at Yangtze River bridges. Therefore, building new bridges should be considered to ease traffic congestion between the south and the north.

### 4.5. Validation and Comparison

4.5.1. Analysis for the Validation of the Result

This study provides a method to mine the overall regularity of traffic congestion in expressway networks from big OD data in terms of temporal changes and spatial disparity. The results about spatiotemporal traffic congestion patterns would be validated by analyzing their rationalities and consistencies with live data in the study region.

The total number of congested locations in four directions was calculated for each day to analyze the rationality of the temporal changes in traffic congestion (Figure 6).

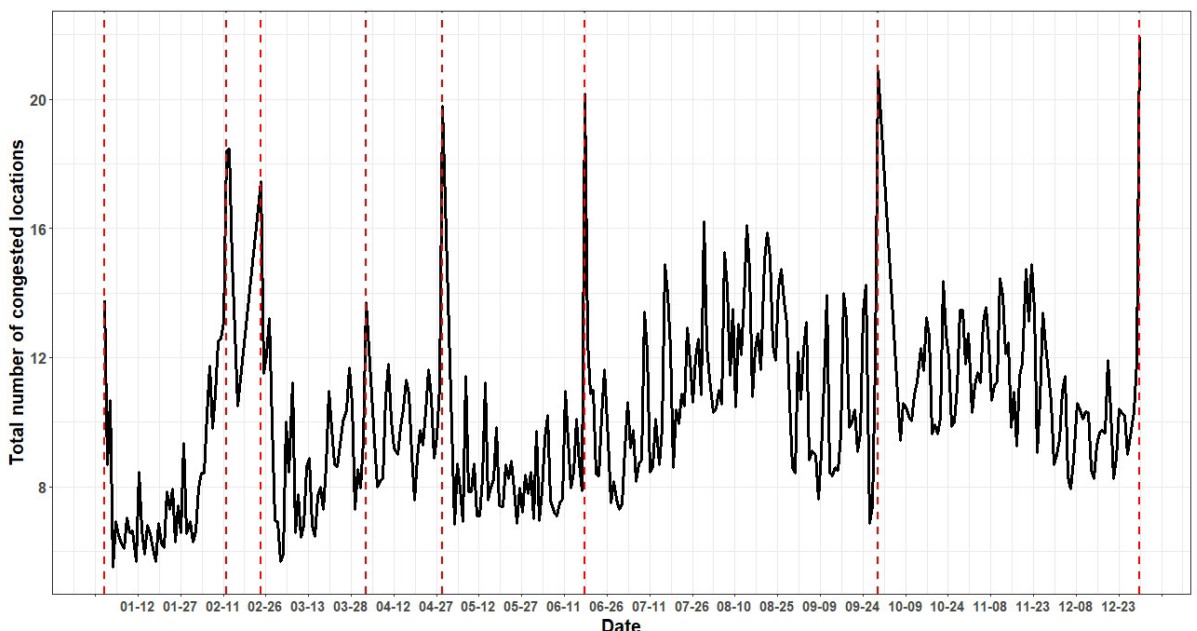

**Figure 6.** Time series of the number of congested locations.

The peak numbers of congested locations that occurred in the expressway network were extracted (Table 5).

**Table 5.** Dates of peak congestion and their descriptions.

| Date of the Peak | Descriptions |
| --- | --- |
| 1 January 2015 | The first day of the New Year's Day Holiday. |
| 13 February 2015 | The last working day before the Chinese New Year Holiday. |
| 25 February 2015 | The first day after the Chinese New Year Holiday. |
| 3 April 2015 | The last day before the Tomb-Sweeping Day Holiday. |
| 30 April 2015 | The last day before International Workers' Day Holiday. |
| 19 June 2015 | The last day before the Dragon Boat Festival Holiday. |
| 30 September 2015 | The last day before the National Day Holiday. |
| 31 December 2015 | The last day before the New Year's Day Holiday. |

The days in Table 5 were all near nationwide holidays in China. In China, traffic congestion occurs on a large spatial scope near nationwide holidays in the expressway network because people choose the expressway to travel. Many peaks were on the last days before the holidays, shown in Table 6, as people tend to start their vacations immediately after work. To summarize, the temporal changes in traffic congestion extracted by the proposed method could reflect real-world circumstances.

**Table 6.** Comparison of the proposed method and other similar methods in the literature.

| Literature | Type of Moving Data | Road | Spatial Range |
| --- | --- | --- | --- |
| Kan et al. [24] | GPS trajectory | Urban road | Turn level |
| Liu et al. [26] | GPS trajectory | Urban road | Road level |
| Zhang et al. [12] | GPS trajectory | Urban road | City |
| Kalinic et al. [32] | GPS data | Expressway | Road level |
| Jianming et al. [37] | Spatiotemporal trajectory | Expressway | Road level |
| This method | OD data | Expressway | Regional |

The spatial patterns in traffic congestion road segments showed that the south and north were more frequently congested due to economic elements and natural hindrances in the study region. It was found that several Yangtze River-spanning bridges in the study region were detected as road segments with frequent congestion (Figure 5), conforming to the idea that river-spanning bridges cause traffic congestion because they are the main passageway to cross the river. Therefore, the spatial patterns of traffic congestion could be in accordance with the study region.

Based on the above analysis, it was found that the temporal and spatial patterns of these traffic congestion could be in accordance with the actual situation in the expressway network of the study region, proving the effectiveness of the proposed method.

### 4.5.2. Compare with Other Similar Methods

This section compares the proposed method with other similar methods in terms of analyzing traffic congestion. The proposed method firstly detects the road segment where traffic congestion happens each hour using OD data. Then, the number of congested locations for each day is calculated, and the frequency of traffic congestion for each road segment is analyzed for the spatiotemporal patterns of traffic congestion in the expressway network.

Table 6 shows the comparison of this method with other similar methods. Other methods mainly utilize GPS trajectory data to analyze congestion on urban roads or partial expressways. GPS trajectory data contains a detailed record of the spatial movement of vehicles, which could detect the accurate location of traffic congestion in other methods. However, GPS trajectory data might be difficult to collect over a whole expressway network because GPS devices need to be installed in each vehicle [24]. Besides, GPS trajectory data cannot be obtained for expressway networks where there is no (or a poor) GPS signal. OD data is an available data type for recording the traffic conditions in regional expressway networks. Due to the lack of trajectory information, most existing studies

ignore traffic congestion data from OD records, and other methods pay less attention to the spatiotemporal patterns of traffic congestion in expressway networks.

By recovering the possible driving routes for OD data, eliminating the influence of the noise, determining the congested route, and selecting the congested road segment from a congested route, this method could be used to analyze the overall spatiotemporal patterns of traffic congestion at the road segment level for regional expressway networks. Compared with other methods for analyzing traffic congestion, this method might be appropriate for regional expressway networks to uncover the spatiotemporal patterns of traffic congestion.

## 5. Discussions and Conclusions

This study proposes a method for utilizing OD data to explore and analyze spatiotemporal patterns of traffic congestion in a regional expressway network. The proposed method selects abnormal OD records with lower-than-average speeds. Based on the proportion of abnormal OD records in OD records, this method detects congestion in expressway networks and identifies the congested road segments in each congested route. Finally, the number of congested locations and the frequency of traffic congestion are calculated to analyze the spatiotemporal patterns of traffic congestion under different directions. Taking expressway networks among 25 cities in Jiangsu Province of China as a case study, this study found that the proposed method could detect the road segments where traffic congestion occurs. The method could also uncover the temporal changes at congested locations and the spatial distribution of road segments with frequent congestion. The spatiotemporal patterns of traffic congestions extracted by this new method could be in accordance with the actual situation of the study region.

This study could not determine the accurate location (longitude and latitude) and distributed length of traffic congestion in expressway networks. The spatiotemporal patterns of traffic congestion were explored and analyzed at the level of road segments. There were 347 road segments in this study, and the average length of road segments was 7.69 km. Therefore, improving positioning accuracy of traffic congestion in the expressway network should be explored in future work. For example, we could infer the distance between the traffic congestion and the expressway exit to indicate the precise location of traffic congestion. The proposed data-based method could explore the spatiotemporal patterns of traffic congestion but could not distinguish between the types of traffic congestion caused by different factors (such as bad weather). Therefore, with the help of other information (such as weather records), classifying and analyzing different factors that cause traffic congestion discovered by the proposed method should be investigated in the future.

The innovation of this method is utilizing OD data to explore and analyze spatiotemporal patterns of traffic congestion in expressway networks. Firstly, existing studies rarely discuss the spatiotemporal patterns of traffic congestion from the perspective of expressway networks because data that cover the whole expressway network are expensive and difficult to collect. On the other hand, existing studies do not use OD data to discuss traffic congestion because OD data lack trajectory information. By recovering the driving routes in OD data, eliminating the influence of the noise, determining the congested route, selecting the congested road segment, and analyzing spatiotemporal patterns of traffic congestion, the proposed method could explore the spatiotemporal patterns of traffic congestion in the expressway network. The contribution of this study was to provide a new approach to uncovering the spatiotemporal patterns of traffic congestion in regional expressway networks. This study might be useful for policymakers and regional managers for optimizing expressway network planning, managing traffic flow in expressway systems, and promoting balanced regional development.

**Author Contributions:** Conceptualization, Linwang Yuan; Data curation, Zhaoyuan Yu; Formal analysis, Hong Gao and Xu Hu; Funding acquisition, Zhaoyuan Yu, Wen Luo, Linwang Yuan and Jiyi Zhang; Methodology, Hong Gao and Jiyi Zhang; Supervision, Linwang Yuan; Visualization, Zhenjun Yan; Writing—original draft, Hong Gao, Zhenjun Yan and Xu Hu; Writing—review and editing,

Zhaoyuan Yu, Wen Luo and Jiyi Zhang. All authors have read and agreed to the published version of the manuscript.

**Funding:** This research was funded by National Natural Science Foundation of China (grant number 41971404, 41976186 and 42001325), National Science Foundation for Distinguished Young Scholars of China (grant number 41625004), and Natural Science Foundation of Anhui Province, China (grant number 2008085QD168).

**Data Availability Statement:** Not applicable.

**Conflicts of Interest:** The authors declare no conflict of interest.

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
