# Peer review of "A Method for Exploring and Analyzing Spatiotemporal Patterns of Traffic Congestion in Expressway Networks Based on Origin–Destination Data"

_ijgi, doi:10.3390/ijgi10050288_

Round 1

Reviewer 1 Report

The authors really improved the paper with respect to the previous version. They answered to all the comments I made in my previous revision.

Author Response

The authors really improved the paper with respect to the previous version. They answered to all the comments I made in my previous revision.

We are very grateful for the previous comments of the reviewer.

Reviewer 2 Report

The authors have improved the paper over the previous version.
There are just a few small things I would like to point out.
I would like to suggest that the authors move the definitions to paragraph 2, 'Basic idea'. In this way the understanding of the text is more immediate.
Secondly, I would like to suggest improving the quality of the figures, because the text is not legible.

Author Response

The authors have improved the paper over the previous version.

We thank the reviewer for the comments on the previous manuscript.

There are just a few small things I would like to point out.

I would like to suggest that the authors move the definitions to paragraph 2, 'Basic idea'. In this way the understanding of the text is more immediate.

Thanks for the valuable suggestions of the reviewer, we moved the related definitions to the section of Basic idea.

Secondly, I would like to suggest improving the quality of the figures, because the text is not legible.

We apologize for the confusion caused by the low-quality figures, and thank for the constructive suggestions of the reviewer. We replaced the existing pictures in the manuscript with high-quality ones.

Reviewer 3 Report

In this study, authors propose a method to uncover the spatio-temporal patterns of traffic congestions in regional expressway network. The paper is generally well organized. However, the methodology part of this paper is weak and some details need to be further clarified. 1. Authors use ‘spatiotemporal patterns’ in the title. However, it seems that authors only discover the place and time of congestions. I don’t think the word ‘spatiotemporal patterns’ is proper for this study. 2. Background. This part is a bit brief and many related studies are not mentioned. Authors divide existing studies into three categories based on the data they used. However, the models are much more important than the data. Thus, a table summarizing and categorizing related studies (comparing data, model, contribution, etc.) may help. 3. Is there any reference for the algorithms you proposed? (tables 1, 2, 5) Actually there algorithms are very simple and easy to understand. So it would be very challenging to tell the contributions of the proposed algorithms. I am hoping the authors can clarify the novel part of your proposed method. 4. The results of your method are quite dependent on the thresholds you use. For example, the speed threshold in Table 1 and the proportion threshold in Table 2. However, I did not see the values of these thresholds in the example. Please give the values. And, should we use different values of threshold in different areas? 5. Please note that all figures in this manuscript are not clear. I can't even make out the numbers and letters. The current version is not acceptable. 6. Of concern is that the results are lackluster. Many results are not surprising. Please highlight some results and make some impressive explanations to show the merits of this study. 7. The writing should be carefully checked and improved. Recommendations for further study should be enriched.

Author Response

In this study, authors propose a method to uncover the spatio-temporal patterns of traffic congestions in regional expressway network. The paper is generally well organized. However, the methodology part of this paper is weak and some details need to be further clarified.

  1. Authors use ‘spatiotemporal patterns’ in the title. However, it seems that authors only discover the place and time of congestions. I don’t think the word ‘spatiotemporal patterns’ is proper for this study.

We thank the reviewer for the insightful comments. The proposed method could use OD data to detect the traffic congestion in different locations and different times. But more than that, according to the detected traffic congestion in different road segments of expressway network and different hours of one year, this study analyzes the spatiotemporal patterns of traffic congestion. On one hand, we characterize the comprehensive situation of the spatial distribution of traffic congestion on the same day, and generate the daily time series of this characteristic to analyze the traffic congestion. On the other hand, we also consider the comprehensive temporal characteristics of traffic congestion in the same road segment, and generate the spatial distribution of this characteristic in the level of the road segment to analyze traffic congestion. The spatial and temporal information of traffic congestion are aggregated as two forms to analyze its comprehensive characteristics in this study. Therefore, we thought that the "spatiotemporal patterns" of traffic congestion in the title might be proper.

  1. Background. This part is a bit brief and many related studies are not mentioned. Authors divide existing studies into three categories based on the data they used. However, the models are much more important than the data. Thus, a table summarizing and categorizing related studies (comparing data, model, contribution, etc.) may help.

We agree with the reviewer that models are more important than the data in terms of detecting and analyzing traffic congestion. According to the valuable comments of the reviewer, we add several important studies about traffic congestion in the section of Introduction and Reference. We also add a table to summarize and categorize the existing studies about traffic congestion in expressway (not included traffic congestion in urban area), from the perspective of data, model, applied range and contribution. This table is in line 58.

  1. Is there any reference for the algorithms you proposed? (tables 1, 2, 5) Actually there algorithms are very simple and easy to understand. So it would be very challenging to tell the contributions of the proposed algorithms. I am hoping the authors can clarify the novel part of your proposed method.

We thank the reviewer for valuable comments and suggestions. These algorithms (Table 2, 3, 6, 7) are the key parts of the proposed method, which have no related reference. Each algorithm includes a series of detailed steps, which could explicitly and systematically show what we do. Besides, these tables can help the readers to implement the proposed method. Therefore, we would like to keep these tables in the manuscript. According to the valuable comment of the reviewer, we revised the content in the section of Discussions and conclusions, to emphasize the innovation and the contribution of the proposed method (line 563-576).

  1. The results of your method are quite dependent on the thresholds you use. For example, the speed threshold in Table 1 and the proportion threshold in Table 2. However, I did not see the values of these thresholds in the example. Please give the values. And, should we use different values of threshold in different areas?

We agree with the reviewer that the threshold is the important parameter in our method, including the speed threshold to select abnormal OD records and the proportion threshold to determine the congested route and the congested road segment. The values of these thresholds are in the last paragraph of the section of Study region and data (line 387-393). And, we add the subtitle for this paragraph to make it easy to be found by the readers.

When this method is applied in different regions, the value of the speed threshold should be set according to the minimum speed limit in the corresponding traffic regulations. The speed threshold should be smaller than the minimum speed limit. The proportion threshold is irrelevant to the spatial region, and the values of the proportion threshold in this study could be used in other regions. 

  1. Please note that all figures in this manuscript are not clear. I can't even make out the numbers and letters. The current version is not acceptable.

We apologize for the use of the low-quality pictures in the previous manuscript. We replaced all pictures in the manuscript with high-quality ones.

  1. Of concern is that the results are lackluster. Many results are not surprising. Please highlight some results and make some impressive explanations to show the merits of this study.

We thank the reviewer for the insightful comments. This study proposes a method that could use OD data to explore and analyze spatiotemporal patterns of traffic congestion in expressway network. As pointed out by the reviewer, the result about temporal changes of the detected traffic congestion might not be surprising, because it is basically in accord with the actual situation and our basic knowledge. However, the spatial disparity of the frequency of traffic congestion with different direction in expressway network might exhibit something interesting and useful, which is in Figure 5. We explain such findings and discuss the usefulness of these results in lines 457-475. We also highlight that the result of this study might be useful for policy makers and regional managers to optimize expressway network planning and promote the regional balanced development in the section of Discussions and conclusions (line 563-576).

  1. The writing should be carefully checked and improved. Recommendations for further study should be enriched.

We would like to thank the reviewer for the careful reading. We checked the manuscript and revised the issues about the writing. Besides, we also add the content to discuss the future work of this study in the section of Discussions and conclusions, which is in line 550-562.

Reviewer 4 Report

Writing and presentation: 
  • Please improve quality of figures (e.g. figure 1), and also provide more detailed captions for the figures in the paper
  • Try to avoid informal writing (e.g., "What's more",
  • Overall, the paper reads well and is easy to follow. But there are sentences that don't read well; example 1: "Therefore, it could be determined whether a route happens traffic congestion". Example 2: "route, thus it should further determine which road segment happened traffic congestion in a route". Example 3: page 7 lines 263-264. Example 4: page 8, lines 306-307. Please do a thorough proofreading to fix these issues. 

Other comments: 

  • The paper focuses on expressway roads, but aren't related literature that focus on other high speed roads which could share similar characteristics with expressways?
  • Why don't you consider the impact of weather conditions on travel time? Doesn't have a significant impact?
  • Something like Fig 2 should come earlier in the introduction section to better convey what an expressway looks like, what are entrance/exit stations, etc. Without a graphical example and just by textual description, it becomes difficult for a reader to follow the paper's logic.
  • if an expressway has certain entrance/exit stations, how could a driver use a detour? This is unclear. Also, if a detour would be more effective to reduce travel time, then a traffic-based shortest-path finder solution would not always pick up the entire expressway, which would make some of the preliminary assumptions in this paper questionable. More clarifications on these would be necessary.
  • I realized formula 1 is not time-agnostic, but authors must clearly indicate this. Otherwise, a reader might think you don't care about the time interval.  
  • Page 6, lines 241-243: what are the observations that back this claim? A potential reader might ask how the authors came to this conclusion?
  • Spatial overlay: it is difficult to imagine what the authors imply by spatial overlay; so it is unclear what does "number of layer" mean in table 3. A simple graphical example could significantly increase readability.
  • Table 4 summarizes four types of travel direction, but how about mixed types? Say South-east for instance?T
  • The density of the case study is mentioned as 700 m/km2, but what does "m" stand for?
  • When using Travel time based on your dataset to obtain congestion segments/roads, do you also consider the potential delay at toll booths when a driver is paying? I didn't see anything in the paper about such a very common delay.
  • Figure 3: it is almost impossible to read anything from the figures, please use high-quality figures.
  • Figure 4: it is not clear what is the selected date range for the plot. More detailed captions would be appreciated.
  • Did you just use 7 days of data in 2015 as your case study? It is not clear if you use more data or just 7 days. 
  • Figure 5: what is the time range for this analysis? Again, please improve caption and image quality.
  • Given the very special design of expressway to collect data for in this study, what are the main reasons for you to prove your study is not biased toward the specific design? And why should a potential reader believe your findings can be applied to other high-speed roads all around the world?
  • Aren't there any other OD-based congestion discovery solutions that you could use as a baseline to compare your proposal against them?

Author Response

Writing and presentation:

Please improve quality of figures (e.g. figure 1), and also provide more detailed captions for the figures in the paper

We apologize for the use of the low-quality pictures in the previous manuscript and thank for the suggestion of reviewer. We replaced all pictures in the manuscript with high-quality ones.

Try to avoid informal writing (e.g., "What's more", Overall, the paper reads well and is easy to follow. But there are sentences that don't read well; example 1: "Therefore, it could be determined whether a route happens traffic congestion". Example 2: "route, thus it should further determine which road segment happened traffic congestion in a route". Example 3: page 7 lines 263-264. Example 4: page 8, lines 306-307. Please do a thorough proofreading to fix these issues.

We appreciate the careful read of the reviewer, and revised the writing issues pointed out by the reviewer. Besides, we checked the whole manuscript to improve the writing.

Other comments:

The paper focuses on expressway roads, but aren't related literature that focus on other high speed roads which could share similar characteristics with expressways?

We are grateful to the insightful comments of the reviewer. The contribution of this study is to propose a method that utilizes OD data to explore and analyze spatiotemporal patterns of traffic congestion in expressway network. The related literature does not contain such method that focuses on traffic congestion in expressway network or uses OD data. On one hand, the related literature rarely discusses the spatiotemporal patterns of traffic congestion from the perspective of expressway network, because it is difficult to obtain the research data that cover the whole expressway network. On the other hand, the related literature doesn't use OD data to discuss traffic congestion, because OD data lacks the information of trajectory. By recovering the driving routes of OD data, eliminating the influence of the noise, determining the congested route, selecting the congested road segment and analyzing spatiotemporal patterns of traffic congestion, the proposed method could use OD data to explore the spatiotemporal patterns of traffic congestion in expressway network, which could be helpful to improve and optimize regional expressway network planning. Therefore, our study might be innovative, compared with related literature. According to the valuable comments of the reviewer, we revised the related content to discuss the advantages and innovations of this study in this version of the manuscript (line 563-577).

Why don't you consider the impact of weather conditions on travel time? Doesn't have a significant impact?

We agree with the reviewer that the weather (such as rainy, foggy and snowy) has much influence on the travel time of vehicles, and may cause the traffic congestion. This study is to propose a method that could utilize OD data to explore and analyze spatiotemporal patterns of traffic congestion in the expressway network. The travel time, which might be influenced by weather conditions and other reasons, would be recorded in OD data. Thus, the proposed method could detect the traffic congestion influenced by weather conditions and other reasons. This study focuses on spatiotemporal patterns of traffic congestion in the expressway network, and ignores to distinguish the reasons that cause traffic congestion. According to the comments of the reviewer, we add the content to illustrate that discussing the causes of traffic congestion, which could be discovered by the proposed method, is an important future work of this study (line 557-562).

Something like Fig 2 should come earlier in the introduction section to better convey what an expressway looks like, what are entrance/exit stations, etc. Without a graphical example and just by textual description, it becomes difficult for a reader to follow the paper's logic.

We appreciate the reviewer for the valuable suggestions. According to the comment of the reviewer, we revised figure 1 (line 163-164) which uses a series of graphical examples to illustrate the basic logic of this study. We also revised the relevant contents near the new figure 1, to explain what the regional expressway network looks like, what is the entrance or exit station, and the framework of the basic idea in this research. These revisions are in line 147-162.

if an expressway has certain entrance/exit stations, how could a driver use a detour? This is unclear. Also, if a detour would be more effective to reduce travel time, then a traffic-based shortest-path finder solution would not always pick up the entire expressway, which would make some of the preliminary assumptions in this paper questionable. More clarifications on these would be necessary.

We would like to thank the reviewer for the insightful comments. This study explores and analyzes the spatiotemporal patterns of traffic congestion in expressway network (with toll system). The entrance and exit stations are fixed toll stations where the drivers need to pay. In the expressway network, there is more than one routes between two toll stations. Most drivers would select the shortest route to drive. Some drivers may not drive along the shortest route, which is a kind of detour. This detour would not be effective when transporting in the expressway network. On one hand, the road classification and traffic regulation would generally be same for different routes in the expressway network. Thus, this detour might obtain few extra advantages of driving. On the other hand, because of the large spatial span of expressways, this detour may need to drive much more extra distance in the expressway network. Therefore, this detour could not reduce the travel time. It might be the mainstream for vehicles to drive along the shortest-path in expressway network. According to the comments of the reviewer, we revised the content (line 147-164) to use the graphical examples to explain the characteristic of the expressway network and the basic logic of our study.

I realized formula 1 is not time-agnostic, but authors must clearly indicate this. Otherwise, a reader might think you don't care about the time interval.

We thank the reviewer for the valuable comments. As pointed out by the reviewer, formula 1 takes the travel time into consideration. Formula 1 is to calculate the proportion of abnormal OD records in each time interval. Abnormal OD records is highly decided by the travel time. We add the content (line 228-232) to emphasize the relationship between formula 1 and the travel time.

Page 6, lines 241-243: what are the observations that back this claim? A potential reader might ask how the authors came to this conclusion?

We thank the reviewer to point out the potential issues of these sentences. This conclusion is based on the theoretical analysis of the proposed method. We want to express that the proposed method could distinguish traffic congestion from other normal situations, according to the important characteristic of traffic congestion. We revised the related paragraph (lines 239-246) to make the expression more accurate and objective.

Spatial overlay: it is difficult to imagine what the authors imply by spatial overlay; so it is unclear what does "number of layer" mean in table 3. A simple graphical example could significantly increase readability.

We appreciate the valuable comments of the reviewer. According to these comments, we revised figure 1 (line 163-164) to utilize a series of graphical examples to illustrate the basic logic of this study. The graphical example of spatial overlay is also in figure 1. Besides, figure 1 could also indicate the role of the spatial overlay in the whole method.

Table 4 summarizes four types of travel direction, but how about mixed types? Say South-east for instance?T

Yes, we summarize four kinds of congested direction for congested road segment, because these directions are most important directions. The congested road segment with mixed type could also be classified as one of these four congestion directions, according to the dominant spatial span of the congested road segment. The details to determine the dominant spatial span and the congested direction are in the lines 292-302. We also revised table 4 in the previous manuscript, which is table 5 in this version, to illustrate the determination of congestion direction for the congested road segment with mixed types. Besides, the expressway like T would be split as three road segments in this method, because road intersection is the breakpoint to divide expressway network into many road segments (line 109-112). Thus, we would not to discuss the congested direction for such situation.

The density of the case study is mentioned as 700 m/km2, but what does "m" stand for?

We thank the comments of the reviewer. We revise this sentence to explain the density of expressway, which means that there is a 700 m expressway per square kilometer in the study region. The related revisions are in line 364-365.

When using Travel time based on your dataset to obtain congestion segments/roads, do you also consider the potential delay at toll booths when a driver is paying? I didn't see anything in the paper about such a very common delay.

We agree with the reviewer that the potential delay at toll booths could also influence the travel time. This study proposes a method that could use OD data to explore and analyze spatiotemporal patterns of traffic congestion in expressway network. The travel time influenced by this potential delay would be recorded in OD data. Thus, our method could explore and analyze the traffic congestion influenced by this potential delay. It should be noted that many reasons would influence the travel time and cause traffic congestion, such as bad weather. OD data would record the travel time that might be influenced by all kinds of reasons. Therefore, the proposed method could explore and analyze traffic congestion caused by many reasons. Discussing many different causes of traffic congestion in expressway network is not the focus of this study. According to the comments of the reviewer, we add the content to illustrate that classifying and analyzing different causes of traffic congestion in the expressway network is an important future work of this study (line 557-562).

Figure 3: it is almost impossible to read anything from the figures, please use high-quality figures.

We apologize for the use of the low-quality pictures in the previous manuscript. We replaced all pictures in the manuscript with high-quality ones.

Figure 4: it is not clear what is the selected date range for the plot. More detailed captions would be appreciated.

The date range in the figure 4 is from January 1, 2015, to December 31, 2015, and each unit means a single day.

Did you just use 7 days of data in 2015 as your case study? It is not clear if you use more data or just 7 days.

No, we use all OD data in the year of 2015 as our case study, to explore and analyze spatiotemporal patterns of traffic congestion in expressway network. The relevant content about the data is introduced in line 387.

Figure 5: what is the time range for this analysis? Again, please improve caption and image quality.

We use all OD data in the year 2015 to investigate the spatiotemporal patterns of traffic congestion in expressway network. Therefore, the time range of figure 5 is the whole year of 2015.

Given the very special design of expressway to collect data for in this study, what are the main reasons for you to prove your study is not biased toward the specific design? And why should a potential reader believe your findings can be applied to other high-speed roads all around the world?

We thank the reviewer for the insightful comments. The proposed method in this study is designed for the expressway network that has the toll system. The toll expressway would record the origin and the destination of vehicle to decide how much to charge, which could collect the OD data. Besides, the toll expressway is a closed system where vehicles must drive in or drive out at fixed toll station. The vehicle can only drive along expressway network between the origin station and the destination station. Thus, we could recover the driving route of OD data to discuss traffic congestion in the expressway network. This kind of toll expressway might not be the specific design, because many other countries also have the toll expressway. Therefore, we think that our method might be available for toll expressway network in other countries.

Aren't there any other OD-based congestion discovery solutions that you could use as a baseline to compare your proposal against them?

Thanks for the valuable comments of the reviewer. After reviewing the literature, we do not find other method that uses OD data to discuss the traffic congestion from the perspective of expressway network. Some similar methods are based on GPS trajectory data to identify and analyze the traffic congestion in the section of expressway instead of expressway network. We compare the proposed method with several similar trajectory-based methods in the section of Validation and comparison (line 509-534), to illustrate the advantages and characteristics of the proposed method.

Round 2

Reviewer 3 Report

Authors made detailed responses to my comments. I suggest acceptance.

Author Response

We thank the reviewer for the valuable comments.

This manuscript is a resubmission of an earlier submission. The following is a list of the peer review reports and author responses from that submission.

Round 1

Reviewer 1 Report

This paper presents an OD record-based approach to investigate the spatial pattern of expressway traffic congestion. The driving path was recovered from the OD. The travelling speed was calculated. The traffic congestion was identified with a speed threshold. The spatial pattern of traffic congestion is calculated. This work is interesting. But there are some problems to be well addressed. 1. Traffic congests in expressway occurs when there is a traffic accident or there is a bottleneck at the exit. Usually, traffic congestion is less than 3 km. There are extreme traffic congestions shown in Figure 10. Therefore, the calculation of driving speed is not right. Generally, the driving speed along the path are different. 2. How to validate the got results? There are many issues should be addressed. For example, the shortest path is not always right. Some drivers may detour for their wrong route choice. For the found traffic congestion, would you please compare it with the online map, like Baidu and Gaode? 3. What is your implication for expressway management? Could you provide some insights for the expressway control with your findings?

Reviewer 2 Report

I think the paper shows an interesting approach on how to describe the traffic flows on the expressways and identify the moments of greatest congestion. In any case, the authors should emphasize more the importance of their study and the innovation in relation to traditional techniques.

In fact, the paper totally misses a section relating to the analysis of the reference literature that could reveal how the methodology proposed by the authors is innovative and is also more efficient than other techniques. Only in the Discussion section the authors provide with reasons for using their method which, however, are not supported by the reference literature.

Without a proper literature analysis the reader is not able to understand if (and why) the methodology is innovative and useful.

An important point that should be addressed is the comparison with techniques which relias on the use of GPS tracking and, more in general, floating car data. This is the most used technology currently and it does not look to be excessively expensive in terms of storage space. Moreover the real advantage of this technique is the opportunity to have real-time data and, consequently, real-time traffic conditions. The authors should therefore highlight in which cases their methodology could be more efficient (e.g. inability to obtain GPS data, rural areas, areas where the signal is missing, post-hoc analysis...)

Finally in discussion/conclusion section the authors should explain to whom and how the proposed methodology could be useful (policy makers, city administrators, transport planners...)

Reviewer 3 Report

The paper deals with an interesting and innovative topic, such as the spatial distribution of traffic congestion.

The authors specifically analyze the spatial distribution of traffic congestion considering expressway network in China.

In my opinion, there are some aspects that should be improved in the paper:

  • In paragraph 2 please explain how express od data are obtained and specify the time interval to which reference will be made to conduct the analysis.
  • In the paragraph 3.3 please specify better the concept of hot spot and cold spot. It is not clear for those unfamiliar with spatial analysis methodology.
  • In paragraph 4 please give some information on the geometric and functional characteristics of a generic expressway.
  • The discussion should be based on the analysis of the results obtained. In its present form, paragraph 5 can be seen as a paragraph of conclusions rather than of discussion.
  • In the conclusions please include proposals for possible practical applications of the results obtained.

Reviewer 4 Report

1,How to define and validate the congestion status from your proposed method? This should be key issues pending clear statement. The current records still need more information to infer traffic congestion. 2, This paper used the Dijkstra algorithm to recover the shortest path between Origin and destination. Do it fit with the dense networks in the study area? Especially, when drivers knew the traffic status of highspeed ways. 3, According to the previous two questions, I am not convinced to accept the results in this paper.